# Mixed-Gas Selectivity Based on Pure Gas Permeation Measurements: An Approximate Model

**DOI:** 10.3390/membranes11110833

**Published:** 2021-10-28

**Authors:** Alexander O. Malakhov, Vladimir V. Volkov

**Affiliations:** A.V. Topchiev Institute of Petrochemical Synthesis, Russian Academy of Sciences, Leninsky Prospect 29, 119991 Moscow, Russia; vvvolkov@ips.ac.ru

**Keywords:** model, gas separation, selectivity, permeability, coupling, polymer membranes

## Abstract

An approximate model based on friction-coefficient formalism is developed to predict the mixed-gas permeability and selectivity of polymeric membranes. More specifically, the model is a modification of Kedem’s approach to flux coupling. The crucial assumption of the developed model is the division of the inverse local permeability of the mixture component into two terms: the inverse local permeability of the corresponding pure gas and the term proportional to the friction between penetrants. Analytical expressions for permeability and selectivity of polymeric membranes in mixed-gas conditions were obtained within the model. The input parameters for the model are ideal selectivity and solubility coefficients for pure gases. Calculations have shown that, depending on the input parameters and the value of the membrane Peclét number (the measure of coupling), there can be both a reduction and an enhancement of selectivity compared to the ideal selectivity. The deviation between real and ideal selectivity increases at higher Peclét numbers; in the limit of large Peclét numbers, the mixed-gas selectivity tends to the value of the ideal solubility selectivity. The model has been validated using literature data on mixed-gas separation of *n*-butane/methane and propylene/propane through polymeric membranes.

## 1. Introduction

Polymeric membranes have been widely used in various gas separation applications mainly due to their high performance regarding permeation and selectivity, as well as the easy processability of polymers. Gas permeation through a dense polymeric membrane is governed by a solution-diffusion mechanism where gas dissolution on the feed side and diffusion across the membranes determine an overall gas separation process. Although single gas permeation properties are more often reported, the selectivity of membranes for mixed gas can be different compared to the ideal selectivity based on single gas measurements [1,2,3,4,5,6,7,8,9]. In some cases, ideal selectivities are higher than mixed gas selectivities. For example, CO_2_/N_2_ selectivities based on single gas measurement might be higher than mixed-gas selectivities due to the swelling of the polymer in CO_2_ environment and plasticization effect [10]. The same tendency was observed for hydrocarbons separation through polyalkylmethylsiloxanes composite membranes [9]. The separation of an eight-component mixture of saturated and unsaturated hydrocarbons C1-C4 was studied. It was shown that the values of ideal selectivities for C_3_H_8_/CH_4_ and *n*-C_4_H_10_/CH_4_ gas pairs were higher than mixed-gas selectivities. This effect was explained by significant swelling of the membrane material in the hydrocarbon mixture, first of all, due to the presence of *n*-butane.

On the other hand, an opposite effect of the increase of the separation selectivity for gas mixtures is a character feature of high free volume glassy polymers (so-called polymers of intrinsic microporosity) [1,7]. This unique effect was first reported for poly(1-trimethylsilyl-1-propyne) (PTMSP) by Srinivasan et al. [11]. It was shown that helium and nitrogen permeation was drastically reduced in the presence of a more strongly sorbing and/or more condensable SF_6_ in the feed; the phenomenon was called “pore blocking” or “light gas rejection” effect. Thus, the ideal selectivity $\alpha_{12}$, calculated by the ratio of the permeabilities for pure gases 1 and 2 can differ rather significantly from the actual, mixed-gas selectivity $\alpha_{12}^{mix}$.

From the theoretical point of view, the deviation of the actual selectivity from the ideal selectivity is due to the complex, interdependent transport of mixture components through the polymer membrane. The reason for the marked reduction or enhancement of membrane selectivity can be designated as the coupling effect. This effect has sorption and diffusion constituents. Sorption coupling is associated with the intermolecular interaction of sorbed species and competition between unlike molecules for the limited number of available “sorption sites” in the polymer. Diffusion coupling is related to friction interaction between the mixture components in the membrane and changes the mobility of the components, slowing the transfer of lighter (faster) gas and accelerating the transfer of heavier (slower) gas.

There is a longstanding, perhaps not entirely correct, opinion that the gas mixture components permeate across rubbery polymers essentially independently of each other. For this reason, pure gas solubilities, diffusivities, and permeabilities can be used in mixed-gas calculations [12]. In the framework of linear irreversible thermodynamics, this means that the cross-terms in the flux differential equations can be neglected, in other words, the coupling of component fluxes can be ignored. Gas permeation properties of rubbery polymers were studied by a combination of the Maxwell–Stefan and Flory–Huggins theories in the works [13,14,15]. It has been obtained that the experimental data on mass transfer of CO_2_/C_2_H_6_ [13,14] and CO_2_/CH_4_ [15] mixtures in polyethylene oxide and polydimethylsiloxane, respectively, can be described with acceptable accuracy without regard to diffusional coupling. On the other hand, for pervaporation and gas separation by rigid glassy polymers, consideration of cross-terms in the flux equations seems necessary and, accordingly, the coupling effect cannot be neglected. It has been found that the friction interactions of penetrant-glassy polymer and penetrant-penetrant are of the same order [16,17].

It should be particularly emphasized that mixed-gas measurements need to modify and improve conventional and widely used experimental techniques [18,19]. Another problem is related to the accurate measurement of gas mixture sorption in polymer materials. This is a non-trivial and time-consuming procedure that has only been implemented for a few polymer membranes [20,21]. According to Genduso et al. [15], this problem represents a “bottleneck in gas transport analysis” in polymer membranes. The limited experimental data on diffusion and sorption of gas mixtures in polymeric membranes, on the one hand, restrain the formulation of rigorous theoretical approaches and, on the other hand, increases the importance of developing engineering predictive models to analyze the permeation properties of polymeric membranes.

The seminal work on the analysis of coupling in pervaporation (the description of pervaporation and gas separation are mathematically identical) was performed by Kedem [22] based on the friction-coefficient formalism. Kedem derived the flux expressions and considered two limiting cases: weak and strong coupling between the penetrant fluxes. However, the application of Kedem’s model to the description of mixed-gas permeation revealed difficulties associated with the interpretation of component permeability coefficients and obtaining the correct expression for the separation factor.

In this work, an attempt was made to develop a simplified analytical model for the evaluation and possibly prediction of mixed-gas transport properties of polymer membranes based on available experimental data on single gases. The basic idea was to split the local permeability into two contributions, one of which depends on the friction between the penetrants, and the other contribution can be identified with pure gas permeability.

## 2. Basic Equations

The Maxwell–Stefan approach (equivalent to the friction-coefficient formalism) based on the concepts of linear irreversible thermodynamics is the classical method for the description of mass transfer [23,24,25]. If the isothermal process in the three-component system, penetrant 1/penetrant 2/polymer 3, is considered, then the Maxwell–Stefan equation (in the absence of external forces and shear forces) for each of the components can be described as follows:(1)$$-\frac{d\mu_{i}}{dz}=\sum_{j=1}^{3}f_{ij}\left(u_{i}-u_{j}\right)$$

This equation defines the balance of the thermodynamic force, namely, the negative gradient of the component chemical potential along the *z* direction perpendicular to the membrane surface, exerted on the molecule of type *i*, and friction forces acting between this component and the molecules of other species. The friction force is the product of the friction coefficient $f_{ij}$ and the difference in velocity between species. The friction coefficient $f_{ij}$ is proportional to the concentration of component *j*, which for polymer systems is conveniently expressed in terms of the volume fraction in the membrane, i.e.,
(2)$$f_{ij}=\phi_{j}\zeta_{ij}=\phi_{j}RT/D_{ij}$$

Here the factor *RT* is the product of the gas constant and absolute temperature, $\zeta_{ij}$ are the mutual friction (resistant) coefficients inversely proportional to the Maxwell–Stefan diffusion coefficients $D_{ij}$. These coefficients are non-symmetric and satisfy the relation $D_{ij}/V_{j}=D_{ji}/V_{i}$, where $V_{i}$ is the pure liquid volume of component *i*. (Note that one can also use the more traditional molar fraction-based diffusivities ${Đ}_{ij}$. The relation between ${Đ}_{ij}$ and $D_{ij}$ is given by ${Đ}_{ij}=D_{ij}\bar{V}/V_{j}$ where $\bar{V}$ is the mean molar volume of the system components).

The average velocity of molecules $u_{i}$ is defined through the molar flux of the component $N_{i}$ (moles of species *i* per membrane unit area per unit time) and its molar concentration $c_{i}$
(3)$$u_{i}=N_{i}/c_{i}$$

With the account for Equations (2) and (3) and assuming $u_{3}=0$ (stagnant polymer matrix), Equation (1) can be rearranged to yield $N_{i}$ as a function of the thermodynamic forces:(4)$$N_{1}=-c_{1}\frac{D_{1}}{RT}\frac{d\mu_{1}}{dz}+c_{1}N_{2}\frac{D_{1}}{D_{12}}V_{2}$$
(5)$$N_{2}=-c_{2}\frac{D_{2}}{RT}\frac{d\mu_{2}}{dz}+c_{2}N_{1}\frac{D_{2}}{D_{21}}V_{1}$$
where
(6)$$D_{1}^{-1}=\frac{\phi_{2}}{D_{12}}+\frac{\phi_{3}}{D_{13}},\;\;D_{2}^{-1}=\frac{\phi_{1}}{D_{21}}+\frac{\phi_{3}}{D_{23}}$$

Here, $D_{i}$ is the local diffusion coefficient of species 1 or 2 in the membrane, $\phi_{i}=c_{i}V_{i}$ is the volume fraction of component *i*. Note that $D_{12}/V_{2}=D_{21}/V_{1}$. In the absence of intermolecular friction between the components 1 and 2, i.e., at $D_{12},D_{21}\to \infty$, Equations (4) and (5) are reduced to
(7)$$N_{i}^{0}=-c_{i}\frac{D_{i}^{0}\;}{RT}\frac{d\mu_{i}}{dz},\;i=1,2$$
where
(8)$$D_{i}^{0}\;=D_{i3}/\phi_{3}$$

A superscript “0” indicates the absence of diffusional coupling between fluid fluxes. The diffusion coefficient $D_{i}^{0}$ referred to a polymer-fixed frame of reference.

Assuming ideal behavior in the gas phase which would be in equilibrium with the sorbed gas at position *z*, i.e., $d\mu_{i}=RT\;d(\ln p_{i})$, Equations (4) and (5) can be rewritten as
(9)$$N_{1}=\Pi_{1}\left(-\;\frac{dp_{1}}{dz}+p_{1}\frac{N_{2}V_{2}}{D_{12}}\right)$$
(10)$$N_{2}=\Pi_{2}\left(-\;\frac{dp_{2}}{dz}+p_{2}\frac{N_{1}V_{1}}{D_{21}}\right)$$
where
(11)$$\Pi_{i}=D_{i}K_{i},\;i=1,\;2$$
is the local permeability coefficient, $K_{i}=c_{i}/p_{i}$ is the local solubility coefficient.

## 3. Kedem’s Solution

Equations (9) and (10) are identical to those obtained by Kedem [22] (in Kedem’s paper the coupling coefficient was written as $\bar{Q}=V_{1}/D_{21}=V_{2}/D_{12}$). In general, permeability coefficients depend on local partial pressures: $\Pi_{i}=\Pi_{i}(p_{1},\;p_{2})$. Assuming that the permeability coefficients are not partially pressure dependent, Equations (9) and (10) can be integrated to give
(12)$$N_{1}l=\Pi_{1}\frac{B_{2}}{1-e^{-B_{2}}}\left(p_{1}^{f}-p_{1}^{p}e^{-B_{2}}\right)$$
(13)$$N_{2}l=\Pi_{2}\frac{B_{1}}{1-e^{-B_{1}}}\left(p_{2}^{f}-p_{2}^{p}e^{-B_{1}}\right)$$
where *l* is the actual thickness of a membrane and
(14)$$B_{1}\equiv \frac{N_{1}lV_{1}}{D_{21}},\;\;B_{2}\equiv \frac{N_{2}lV_{2}}{D_{12}}$$

Without coupling ($B_{1},B_{2}\to 0$) Equations (12) and (13) are reduced to
(15)$$N_{i}l=\Pi_{i}\left(p_{i}^{f}-p_{i}^{p}\right)$$
where $p_{i}^{f}$ and $p_{i}^{p}$ are the partial pressures of the type *i* component in feed and permeate, respectively. In a strongly coupled system ($B_{1},B_{2}\geq 5$)
(16)$$N_{1}l=\Pi_{1}p_{1}^{f}B_{2}$$
(17)$$N_{2}l=\Pi_{2}p_{2}^{f}B_{1}$$

Taking into account Equation (14), from the ratio of Equations (16) and (17) it follows that
(18)$${\left(y_{1}/y_{2}\right)}^{2}=\Pi_{1}x_{1}/\left(\Pi_{2}x_{2}\right)$$
where $y_{i}=N_{1}/\left(N_{1}+N_{2}\right)$ is the molar fraction of component *i* in the permeate, $x_{i}=p_{i}^{f}/p^{f}$ is the molar fraction of component *i* in feed gas mixture, $p^{f}$ is the total feed pressure.

A basic quantity of interest is the separation factor (SF) for a binary gas mixture defined as
(19)$$\beta_{12}\equiv \frac{y_{1}/y_{2}}{x_{1}/x_{2}}$$

In terms of $\beta_{12}$, Equation (18) reduces to $\beta_{12}=\sqrt{\Pi_{1}x_{2}/\left(\Pi_{2}x_{1}\right)}$. This relation includes permeability coefficients, which, according to Kedem’s assumption, are constant. For the model to be practically usable, permeability $\Pi_{i}$ should preferably be identified with a parameter available from the experiment, such as the permeability of the pure component, and the permeability ratio $\Pi_{1}/\Pi_{2}$ should be identified with the ideal selectivity ($\alpha_{12}$). With this interpretation, the SF is expressed as
(20)$$\beta_{12}\overset{?}{\to }\sqrt{\alpha_{12}x_{2}/x_{1}}$$

However, this expression, apart from the fact that it diverges at $x_{1}\to 0$ and tends to zero at $x_{1}\to 1$, is incompatible with Kedem’s important result for a strongly coupled system. This result (for negligible pressures on the permeate side) was indicated by Kedem herself and is as follows
(21)$$N_{1}/N_{2}\to c_{1}/c_{2},\;\mathrm{hence}\;\beta_{12}\to K_{1}/K_{2}$$

This conclusion can be obtained from the differential flux equation for one of the components. Let us consider Equation (4). In the strong coupling limit ($D_{12}\to 0$) the first term in the right-hand side of this equation can be neglected. Moreover, as follows from Equation (6), the diffusion coefficient $D_{1}\to D_{12}/\left(c_{2}V_{2}\right)$ in this limit, which leads to relation (21). According to the relation, in the limit of strong coupling, the SF tends to the ratio of the solubility coefficients. This inconsistency is evident in Equations (20) and (21). The reason for the discrepancy in the results for $\beta_{12}$ lies in Kedem’s assumption concerning the constancy of the permeability coefficients when integrating Equations (9) and (10). A proposed approach, applying a partitioning of the inverse permeability coefficients into a term explicitly dependent on the intermolecular friction of the penetrants and a term determined by frictional interaction of the penetrant with the polymer matrix, is presented below.

## 4. Model Development

### 4.1. Approximations and Flux Equations

Now, let us return to the differential flux Equations (9) and (10), and consider the permeability coefficients $\Pi_{i}$ (Equation (11)). Substituting Formulas (6) into Equation (11) gives the following expressions for the local permeability coefficients:$$\frac{1}{\Pi_{1}}=\frac{1}{K_{1}}\left(\frac{\phi_{2}}{D_{12}}+\frac{\phi_{3}}{D_{13}}\right),\;\;\frac{1}{\Pi_{2}}=\frac{1}{K_{2}}\left(\frac{\phi_{1}}{D_{21}}+\frac{\phi_{3}}{D_{23}}\right)$$

Since the volume fraction of type *i* penetrant $\phi_{i}=c_{i}V_{i}=K_{i}p_{i}V_{i}$, the above equations become:(22)$$\frac{1}{\Pi_{1}}=\frac{K_{2}}{K_{1}}p_{2}\frac{V_{2}}{D_{12}}+\frac{1}{\Pi_{1}^{0}}$$
(23)$$\frac{1}{\Pi_{2}}=\frac{K_{1}}{K_{2}}p_{1}\frac{V_{1}}{D_{21}}+\frac{1}{\Pi_{2}^{0}}$$
where
(24)$$\Pi_{1}^{0}=K_{1}D_{13}/\phi_{3}=K_{1}D_{1}^{0},\;\;\Pi_{2}^{0}=K_{2}D_{23}/\phi_{3}=K_{2}D_{2}^{0}$$

In the absence of coupling ($D_{12},D_{21}\to \infty$), the first term in the right-hand side of Equations (22) and (23) vanishes, and the permeability coefficients $\Pi_{i}$ are reduced to the coefficients $\Pi_{i}^{0}$, which, as a first approximation, can be assumed to be independent of concentration and identified with the corresponding values for single gases. The relations (22) and (23) play a crucial role in the model development.

Substituting Equations (22) and (23) into Equations (9) and (10), respectively, give
(25)$$\frac{N_{1}l}{\Pi_{1}^{0}}=-\frac{dp_{1}}{d\xi }+p_{1}B_{2}-\frac{K_{2}}{K_{1}}p_{2}B_{1}$$
(26)$$\frac{N_{2}l}{\Pi_{2}^{0}}=-\frac{dp_{2}}{d\xi }+p_{2}B_{1}-\frac{K_{1}}{K_{2}}p_{1}B_{2}$$
where ξ=z/l is dimensionless membrane thickness, the parameters $B_{i}$ are defined in Equation (15). These equations include the ratio of the local solubility coefficients. We make a simplifying assumption $K_{1}/K_{2}=\mathrm{const}$ that allows an analytical integration of the above differential equations. The assumption of constant $K_{1}/K_{2}$ is fulfilled when the sorption of the mixture is consistent with the Henry model, as well as with the multicomponent Langmuir model. It seems reasonable to equate the ratio $K_{1}/K_{2}$ with the ratio of solubility coefficients for pure gases $\alpha_{12,\;S}\equiv \alpha_{S}$, i.e.,
(27)$$K_{1}/K_{2}=\alpha_{S}$$

Identification (27) means that, within the framework of this work, the deviation of the membrane selectivity from ideal selectivity is due only to frictional interaction of the gas penetrants 1 and 2. Namely, the coupling effect is based on the diffusional coupling only. In practice, there may be noticeable deviations ${K_{1}/K}_{2}$ from the ideal solubility selectivity (see, e.g., [20,26]). Given that mixed-gas solubility data are far from always available and their experimental measurement is a time-consuming procedure, the approximation (27) is reasonably adequate for design calculations.

Using Equation (27) and the relation $B_{i}=y_{i}B$, Equations (25) and (26) can be rewritten as
(28)$$\frac{N_{1}l}{\Pi_{1}^{0}}=-\frac{dp_{1}}{d\xi }+B\left(p_{1}y_{2}-p_{2}y_{1}/\alpha_{S}\right)$$
(29)$$\frac{N_{2}l}{\Pi_{2}^{0}}=-\frac{dp_{2}}{d\xi }+B\left(p_{2}y_{1}-\alpha_{S}p_{1}y_{2}\right)$$
where
(30)$$B=\frac{\left(N_{1}+N_{2}\right)lV_{1}}{D_{21}}=\frac{NlV_{1}}{D_{21}}$$
is the quantity that includes the total molar flux *N* of the gas mixture and 1–2 diffusivity $D_{21}$, which is a measure of coupling. In the membrane literature, a dimensionless quantity of this type is often referred to as the “membrane Peclét number”.

Integration of the differential equation system (28) and (29) leads to the result (see Appendix A for some details):(31)$$N_{1}l=\Pi_{1}^{0}\left(\Delta p_{1}+d_{1}\right)$$
(32)$$N_{2}l=\Pi_{2}^{0}\left(\Delta p_{2}-\alpha_{S}\;d_{1}\right)$$
where
(33)$$\begin{array}{l} d_{1}=p^{f}\omega \left(y_{2}\;x_{1}-y_{1}\;x_{2}/\alpha_{S}\right)+p^{f}r(B-\omega )y_{1}y_{2}\left(1-1/\alpha_{S}\right)\\ \omega \equiv \frac{B}{1-\exp(-B)}-1 \end{array}$$

Here *r* is the pressure ratio $r=p^{p}/p^{f}$ ($p^{f}$ and $p^{p}$ denote the feed and permeate pressure, respectively). One can see that besides the partial pressure drop, the flux Equations (31) and (32) include an additional contribution (“coupling pressure”) due to diffusion coupling. Note that this additional contribution has different signs for components 1 and 2, i.e., the coupling results in lower flux (compared to pure gas flux) for one component and higher flux for the other component. This is the apparent difference between the developed model and Kedem’s approach, in which the additional “driven force” (complementary to $\Delta p_{i}$) has a positive sign for both components (it is easy to show from the flux equations (10)). In the limit B→0 (no coupling), Equations (31) and (32) are reduced in form to Equation (15). However, unlike Equation (15), these equations include permeability coefficients which are identified with corresponding quantities for individual gases.

It follows from Equations (31) and (32) that
(34)$$\frac{N_{1}l}{\Pi_{1}^{0}}+\frac{N_{2}l}{\Pi_{2}^{0}}\frac{1}{\alpha_{S}}=\Delta p_{1}+\Delta p_{2}/\alpha_{S}$$

Using the equality $N_{i}=y_{i}N$, one gets an expression for the total flux
(35)$$Nl=\Pi_{2}^{0}\frac{\Delta p_{1}\;\alpha_{12}+\Delta p_{2}\;\alpha_{D}}{y_{1}+y_{2}\;\alpha_{D}}$$
where
(36)$$\alpha_{12}=\Pi_{1}^{0}/\Pi_{2}^{0},\;\;\alpha_{D}=\alpha_{12}/\alpha_{S}$$

In the above equations $\alpha_{12}$ is the ratio of permeability coefficients for pure gases (the ideal selectivity) and $\alpha_{D}$ is the ideal diffusivity selectivity. (By the way, note that if the penetrants have the same mobility, i.e., $\alpha_{D}=1$, then, as seen from (35), the total flux coincides with the total flux of the gas mixture without coupling. This fact is a direct consequence of assumption (27)).

Any two of the three Equations (31), (32) and (35) can be used to obtain the equation for calculating the permeate composition. So, dividing Equation (31) by Equation (32) gives
(37)$$\frac{y_{1}}{y_{2}}=\alpha_{12}\frac{\Delta p_{1}+d_{1}}{\Delta p_{2}-\alpha_{S}\;d_{1}}$$
where $\Delta p_{i}=p^{f}(x_{i}-ry_{i})$. After solving this equation, the SF can be calculated.

### 4.2. Separation Factor

First, consider the trivial case of no coupling, i.e., B=0. In this situation $d_{1}=0$ and Equation (37) reduces to the quadratic equation with the familiar result for the permeate composition [27]. Using the definition (19), the SF for a binary gas mixture with no coupling is given by
(38)$$\begin{array}{l} \beta_{12}^{0}=\frac{1}{2x_{1}}\left(b+\sqrt{b^{2}+4x_{1}x_{2}\alpha_{12}}\right)\;\\ b\equiv \alpha_{12}x_{1}-x_{2}-r(\alpha_{12}-1) \end{array}$$
with ${\beta_{12}^{0}|}_{x_{1}=0}=\alpha_{12}/\left[1+r(\alpha_{12}-1)\right]$. It is evident that $\beta_{12}\leq \alpha_{12}$ (equal sign for *r* = 0).

Next, consider the situation of coupling between component fluxes (B>0). If the downstream pressure is very small, then Equation (37) reduces formally to the quadratic equation with the following solution for the SF:(39)$$\begin{array}{l} \beta_{12}=\frac{1}{2x_{1}}\left(g+\sqrt{g^{2}+4x_{1}x_{2}\alpha_{12}}\right)\\ g\equiv \frac{1-e^{-B}}{B}\left(\alpha_{D}-1\right)\left(x_{1}\alpha_{S}+x_{2}\right)+x_{1}\alpha_{S}-x_{2}\alpha_{D} \end{array}$$

It should be noted that this equation is indeed quadratic if one considers the Peclét number *B* as the *parameter* that does not depend on the permeate composition. In general, this is not the case (the Peclét number is a function of permeate composition, see Section 4.4 below). Nevertheless, Equation (39) is useful as an approximate starting point for further model development. In the case of nonzero downstream pressure (0<r<1), Equation (37) is reduced to a cubic equation for the SF, which is more convenient to solve numerically.

Equation (39) implies that: (1) the SF is finite over the entire concentration range of the feed mixture (in distinction to expression (20)), in the limit $x_{1}\to 0$ one has ${\beta_{12}|}_{x_{1}=0}=\alpha_{12}{\left[1+\left(\alpha_{D}-1\right)\left(1-\left(1-e^{-B}\right)/B\right)\right]}^{-1}$; (2) $\beta_{12}=\alpha_{S}$ if $\alpha_{D}=1$, i.e., for (hypothetical) penetrants with the same diffusion mobility the SF is equal to the solubility selectivity; (3) $\beta_{12}=\alpha_{S}$ also if the Peclét number is much greater than one; (4) in the absence of coupling (B=0) the SF is equal to the ideal selectivity. The last two facts determine the range of SF values:(40)$$\beta_{12}=\left\{ \begin{array}{l} \alpha_{12},\;B=0\\ \alpha_{S},\;B\gg 1 \end{array}\right.\;\mathrm{or}\;\frac{\beta_{12}}{\alpha_{12}}=\left\{ \begin{array}{ll} 1, & B=0\\ \alpha_{D}^{-1}, & B\gg 1 \end{array}\right.$$

An important qualitative conclusion follows from (40): the coupling effect is the *negative* for the diffusivity selectivity membranes ($\alpha_{D}>1$), i.e., real selectivity (mixture selectivity) is less than the ideal one. On the contrary, the coupling effect is positive for the solubility selectivity membranes ($\alpha_{D}<1$), i.e., real selectivity is greater than the ideal one.

### 4.3. Component Permeabilities

The permeability coefficient of the mixture component *i* and mixed-gas selectivity are defined as
(41)$$\Pi_{i}^{mix}\equiv N_{i}l/\Delta \;p_{i},\;\alpha_{12}^{mix}\equiv \Pi_{1}^{mix}/\Pi_{2}^{mix}$$

From Equation (35) for the total flux, the component fluxes are
(42)$$N_{1}l=\Pi_{2}^{0}y_{1}\frac{\Delta p_{1}\;\alpha_{12}+\Delta p_{2}\;\alpha_{D}}{y_{1}+y_{2}\;\alpha_{D}},\;\;N_{2}l=\Pi_{2}^{0}y_{2}\frac{\Delta p_{1}\;\alpha_{12}+\Delta p_{2}\;\alpha_{D}}{y_{1}+y_{2}\;\alpha_{D}}$$

With the relations $y_{1}\;=\;\beta_{12}x_{1}/(x_{2}\;+\;\beta_{12}x_{1})$
and $y_{2}=1-y_{1}$, one gets
(43)$$N_{1}l=\Pi_{1}^{0}x_{1}\;\beta_{12}\frac{\Delta p_{1}\;+\Delta p_{2}/\alpha_{S}}{x_{1}\;\beta_{12}+\;x_{2}\;\alpha_{D}},\;\;N_{2}l=\Pi_{2}^{0}x_{2}\;\alpha_{12}\frac{\Delta p_{1}\;+\Delta p_{2}/\alpha_{S}}{x_{1}\;\beta_{12}+\;x_{2}\;\alpha_{D}}$$

As pointed out above in Equation (40), the SF in the limit of large Peclét numbers is equal to the solubility selectivity. Using this result in Equation (43) and for simplicity, setting *r* = 0, the *limiting* permeability coefficients are
(44)$$\Pi_{1}^{mix}=\Pi_{1}^{0}\frac{x_{1}\;\alpha_{S}+x_{2}}{x_{1}\;\alpha_{S}+\;x_{2}\alpha_{D}},\;\;\Pi_{2}^{mix}=\Pi_{2}^{0}\;\alpha_{D}\frac{x_{1}\;\alpha_{S}+x_{2}}{x_{1}\;\alpha_{S}+\;x_{2}\alpha_{D}}$$

This suggests that $\alpha_{12}^{mix}\equiv \Pi_{1}^{mix}/\Pi_{2}^{mix}=\alpha_{S}$ in the case of a strongly coupled system. It can be shown (see Appendix B) that the expressions (44) are equivalent to the relations previously obtained by Krishna and van Baten [28] in the model description of gas mixture permeation across micro- and mesoporous membranes. 

The implications of the expressions (44) are obvious. If the preferentially permeating component (component 1) of the gas mixture is more mobile, i.e., $\alpha_{D}>1$, then its permeability $\Pi_{1}^{mix}$ will be lower compared to the pure gas permeability $\Pi_{1}^{0}$; the mixed-gas permeability of component 2 will, on the contrary, be higher than in pure gas conditions. If the preferentially permeating component of the gas mixture is less mobile, i.e., $\alpha_{D}<1$, then under conditions of a gas mixture it penetrates through the membrane faster than in a pure state; the second, more diffusion-mobile component, penetrates slower compared to corresponding pure gas. The situation described is reminiscent of Le Chatelier’s principle in chemical thermodynamics. To summarize, one can write
(45)$$\begin{array}{l} \Pi_{1}<\Pi_{1}^{0},\;\;\Pi_{2}>\Pi_{2}^{0}\;\;\mathrm{if}\;\alpha_{D}>1\\ \Pi_{1}>\Pi_{1}^{0},\;\;\Pi_{2}<\Pi_{2}^{0}\;\;\mathrm{if}\;\alpha_{D}<1 \end{array}$$

Note that conditions (45) are obtained for the situation of strong coupling (large Peclét numbers) and assuming downstream pressure is very small. Calculations show that inequalities (45) are also valid under less stringent constraints.

Examples of the SF and component permeability calculations are shown in Figure 1 and Figure 2, respectively. Consider first the case of a sorption-selective membrane, for which $\alpha_{D}<1$ and $\alpha_{S}>\alpha_{12}>1$. The SF increases with the degree of coupling (i.e., the Peclét number) (Figure 1a) and tends to the value of the ideal solubility selectivity. If the pressure ratio *r* = 0, then the SF exceeds the value of ideal selectivity $\alpha_{12}$ (=15 in this example). At non-zero permeate pressure, the SF also grows monotonically with increasing the Peclét number. In this case, two intervals can be distinguished: at small Peclét numbers (*B* less than about 1) $\beta_{12}<\alpha_{12}$, and at appreciable Peclét numbers (*B* more than about 1) the SF even exceeds the corresponding parameter at zero permeate pressure. In the case of a diffusion-selective membrane ($\alpha_{D}∼\alpha_{12}>1$ and $\alpha_{S}∼1$), the SF is less than the ideal selectivity and decreases with an increase in the Peclét number (Figure 1b). In addition, an increase in downstream pressure (from 0 to $0.1p^{f}$) leads to an additional decrease in the SF.

The behavior of the SF will become clearer when considering the dependence of relative permeability coefficients ${\tilde{\Pi }}_{i}=\Pi_{i}^{mix}/\Pi_{i}^{0}$ on Peclét numbers (Figure 2). For a sorption-selective membrane, component 1 is less mobile than component 2 ($\alpha_{D}<1$). As evident from Figure 2a, the mixed-gas permeability of component 1 exceeds pure gas permeability, while the mixed-gas permeability of component 2, on the contrary, is lower than its permeability in pure gas conditions. Namely, slow gas 1 moves faster in the mixture, and fast gas 2, on the contrary, reduces its diffusion mobility. Everything occurs according to the above inequality (45) at $\alpha_{D}<1$. Calculations have shown (dashed lines in Figure 2a) that an increase in downstream pressure leads to a larger deviation of the permeability coefficients from the ideal (pure gas) values. In the case of a diffusion-selective membrane, mixed-gas permeabilities behave in the opposite way to the case of a sorption-selective membrane: the permeability of fast component 1 decreases, and the permeability of slow component 2 increases with Peclét’s number (Figure 2b). At the same time, an increase in downstream pressure has almost no effect on the mixed-gas permeabilities values.

### 4.4. Explicit Form of the Peclét Number

Above, the SF was estimated as a function of the Peclét number using the ideal selectivity and solubility selectivity as input parameters. The effect of coupling was treated formally, in the sense that the Peclét number was assumed to be a parameter independent of feed and permeate composition. A more rigorous approach, not limited to the description of experimental data, should take this dependence into account and provide an explicit expression for the Peclét number. In addition, it is important to evaluate the range of Peclét numbers characteristic of mixed-gas transport through sorption- or diffusion-selective membranes.

Let us now rewrite Equation (30) for the Peclét number as the following:(46)$$B=\frac{N\;l}{\Pi_{2}^{0}p^{f}}p^{f}V_{1}\frac{D_{2}^{0}K_{2}}{D_{21}\;}=\tilde{N}\;p^{f}V_{1}\frac{D_{2}^{0}K_{2}}{D_{21}\;}$$
where $\tilde{N}$ is the dimensionless total permeate flux. The explicit expression of $\tilde{N}$ follows from Equation (35):(47)$$\tilde{N}=\frac{(x_{1}-ry_{1})\;\alpha_{12}+(x_{2}-ry_{2})\;\alpha_{D}}{y_{1}+y_{2}\;\alpha_{D}}$$

The expression for *B* contains the unknown Maxwell–Stefan diffusivity $D_{21}$ which is an inverse friction coefficient between penetrants 1 and 2. Given that Maxwell–Stefan diffusivities are not accessible from the experiment, various interpolation relations are used to predict those [29]. In this paper the simple (geometric mean) approximation is used: $D_{12}D_{21}\approx D_{1}^{0}D_{2}^{0}$ where $D_{i}^{0}$ is the diffusion coefficient for the pure component *i*. Since $D_{12}=D_{21}V_{2}/V_{1}$ then
(48)$$D_{21}\approx {\left(D_{1}^{0}D_{2}^{0}\;V_{1}/V_{2}\right)}^{1/2}$$

Note that this expression for 1–2 diffusivity is similar (and even simpler) to the interpolation formula used by Krishna [14] when modeling water/ethanol pervaporation across a polyimide membrane. Substituting of Equation (48) into Equation (46) gives
(49)$$B=\tilde{N}\;\frac{\sqrt{V_{1}V_{2}}}{22414}\frac{S_{2}p^{f}}{\sqrt{\alpha_{D}}\;}k$$
where $S_{2}$ is the experimental solubility coefficient of component 2 in units cm^3^(STP)/(cm^3^·atm), the feed pressure expressed in atm, a correction factor k∼1 was introduced into the equation to compensate for the approximate nature of equality (48).

Let us estimate the value of *B* depending on the composition of the mixture to be separated. For the sake of simplicity, let the pressure ratio r=0. It follows from Equation (47) that the total dimensionless flux varies from 0 to $\alpha_{12}$ when the mixture composition changes from $x_{1}=0$ (the pure component 2) to $x_{1}=1$ (the pure component 1). According to Equation (49), the value of *B* increases with the upstream pressure and molar volumes of penetrants. At fixed upstream pressure Equation (49) becomes
(50)$$B=\left\{ \begin{array}{l} \mathrm{const}\frac{S_{2}}{\sqrt{\alpha_{D}}\;},\;\;x_{1}\to 0\\ \mathrm{const}\frac{S_{2}\;\alpha_{12}}{\sqrt{\alpha_{D}}\;},\;x_{1}\to 1 \end{array}\right.$$

For instance, if to take $p^{f}=10$ atm, $\sqrt{V_{1}V_{2}}\approx 85$ cm^3^/mol, and k=1, then const≈0.04. Since component 1 is the preferentially permeating component ($\alpha_{12}>1$), the coupling effect is maximum in the limit of infinite dilution of component 2 in the feed mixture. In addition, as can be seen from Equation (50), the value of *B* decreases with increasing diffusivity selectivity. Therefore, one would expect the coupling effect to be weaker for membranes with diffusion selectivity than for membranes with sorption selectivity.

Diffusivity selectivity glassy polymers (polyimides, microporous polymers), which are used to separate olefins and paraffins C_2+_, have almost no sorption selectivity ($\alpha_{S}∼1$), and the ideal selectivity values are $\alpha_{12}∼\alpha_{D}=2-30$ [8,30]. If one accepts $\alpha_{12}=\alpha_{D}=10$ and $S_{2}=10$ cm^3^(STP)/(cm^3^·atm) (as the average solubility of propane in aromatic polyimides) for a rough estimate, then from Equation (50) one obtains that the Peclét numbers vary from 0.2 to 1.9. For solubility selectivity glassy polymers (polyacetylenes, polynorbornenes, PIM-1) used for C_2+_/methane separations, the diffusion selectivity is less than 1, and the sorption selectivity is in a wide range from 5 to ~10^3^ [8,31]. For typical values $\alpha_{12}=10$, $\alpha_{D}=0.2$ and $S_{2}=4$ cm^3^(STP)/(cm^3^·atm) (as the solubility of methane in PTMSP at 25 °C) one gets from Equation (50) the range of Peclét numbers from 0.4 to 3.6. Thus, it can be stated that in the hydrocarbon separation by glassy polymers, the Peclét number reaches values of the order of unity (notice that the above estimates were made at a total feed pressure of 10 atm).

Let us list the basic equations for the mixed-gas selectivity and permeability calculation within the framework of the developed model. Permeate composition is calculated using Equation (37) with the expression (49) for the Peclét number. The relationship of permeate composition with the SF is given by Equation (19). Penetrant fluxes are then calculated using Equations (31) and (32) or, equivalently, via the expressions (42). For negligible pressures of the permeate side, computations are simplified. In this case, the SF can be found by Formula (39). In doing so, the Peclét number can be considered in the first approximation as a model parameter, or it can be estimated using the explicit expression (49). Input parameters for the proposed model are the pure gas permeabilities and solubility coefficients.

### 4.5. Comparison of the Model with Experimental Data

The results of the model calculations were validated against the experimental data for two examples of hydrocarbon separation by the glassy polymers. The first example is *n*-butane/methane separation through high free volume disubstituted polyacetylene—poly(1-trimethylsilyl-1-propyne) (PTMSP). The second example is propylene/propane separation through polyimide 6FDA-TrMPD. These examples illustrate the transport properties of sorption- and diffusion-selective membranes, respectively. Pure gas values of selectivity and solubility coefficients were used as input parameters for calculations (Table 1). Results of the model calculations and their comparison with mixed-gas experimental data are presented in Figure 3.

In the case of PTMSP, the *n*-C_4_H_10_/CH_4_ selectivity is noticeably greater than the ideal value (=3.5) and grows with *n*-butane concentration in feed (Figure 3a (top)). The component permeabilities (in units of pure gas values), on the contrary, decrease with the increasing concentration of *n*-butane (Figure 3a (bottom)). Light gas CH_4_ permeability in mixed-gas conditions decreases as compared to the pure gas value ($\Pi_{{\mathrm{CH}}_{4}}^{mix}<\Pi_{{\mathrm{CH}}_{4}}^{0}$), whereas the situation is the opposite for heavier *n*-C_4_H_10_ ($\Pi_{C_{4}H_{10}}^{mix}>\Pi_{C_{4}H_{10}}^{0}$). This behavior is consistent with the second inequality (45).

In the case of polyimide 6FDA-TrMPD, the mixed-gas C_3_H_6_/C_3_H_8_ selectivity is almost half that of pure gas selectivity and decreases with increasing concentration of preferentially permeating component (propylene) in feed (Figure 3b (top)). The permeability of both components increases with the propylene concentration in the feed (Figure 3b (bottom)). Permeability behavior in mixed-gas conditions follows the first inequality (45): for more mobile gas C_3_H_6_ permeability decreases compared to pure gas one, while for less mobile gas C_3_H_8_ there is an increase in permeability compared to a pure gas one.

Generally, one can state a semi-quantitative agreement with the experiment for both penetrants/polymer systems. As can be seen in Figure 3, the agreement between the calculated and experimental selectivities improves at non-zero permeate pressure. The calculation results depend on the Peclét number. It was evaluated using Equation (49) with fixed parameters $p^{f}=10$ atm, $\sqrt{V_{1}V_{2}}=85$ cm^3^/mol, and k=1. Given the crude approximation (48) for 1–2 diffusivity, the developed model shows more than satisfactory performance.

In closing, it should be pointed out the following. It has been known since [35] that the mixed-gas *n*-C_4_H_10_/CH_4_ selectivity of PTMSP membranes significantly exceeds the ideal selectivity, by a factor of 6–18 according to various experimental data [31]. The increase in selectivity is associated with a marked (3–9 times) decrease in CH_4_ permeability under mixed-gas conditions compared to pure gas conditions [31]. The physical reason for depressing the CH_4_ permeability is commonly attributed to a blocking mechanism: light gas CH_4_ diffusion in free volume elements of ”microporous” PTMSP is hindered, blocked by the sorbed molecules of heavier, condensable *n*-C_4_H_10_ [11,35]. Along with this basic reason, one should also take into account the effect of competitive sorption, which can lead to a change (increase) of solubility selectivity [20,33]. In terms of the developed model, the blocking of methane transport by *n*-butane is a consequence of frictional interaction between penetrants, in other words, the blocking mechanism is a manifestation of coupling between methane and *n*-butane fluxes.

## 5. Conclusions

An approximate model for the mixed-gas selectivity and permeability of non-porous and microporous polymeric membranes was developed. This model is the modification of Kedem’s friction-coefficient approach [22] to flux coupling in pervaporation. The central argument of the model is to decompose the inverse local permeability of penetrants into two terms. The first one is the inverse permeability of pure gas, and the second one is proportional to the friction between penetrants. As a result, under certain simplifying assumptions (constant values of local pure gas permeability and solubility selectivity), analytical expressions for permeability and selectivity of polymeric membranes in mixed-gas conditions were derived. It is shown that the use of simple (geometrical mean) approximation for the Maxwell–Stefan penetrant 1/penetrant 2 diffusivity allows considering the developed model as a predictive one if pure gas permeability and solubility coefficients are available as input. It was found that the coupling between penetrant fluxes causes the membrane selectivity to deviate from pure gas selectivity. For diffusion-selective membranes, there is a reduction in selectivity, while for sorption-selective membranes there is an enhancement in selectivity with respect to pure gas selectivity. A quantitative measure of the coupling effect in the model is the membrane Peclét number. As estimates have shown, the Peclét number is less or of the order of unity in hydrocarbon separation by glassy polymers. In the limit of large Peclét numbers, the derived expressions have the same final form as those obtained earlier by Krishna and van Baten [28] in modeling the permeability of gas mixtures across the inorganic microporous materials. In this limiting case, the mixed-gas selectivity is equal to the value of the ideal solubility selectivity.

The model was tested for the separation of *n*-butane/methane and propylene/propane mixtures through PTMSP and polyimide 6FDA-TrMPD membranes, respectively. These two examples illustrate the hydrocarbon separation with sorption-selective (PTMSP) and diffusion-selective (6FDA-TrMPD) membranes. A rough quantitative agreement of the model results with the experimental data was obtained. Thus, the model evaluation of mixed-gas selectivity using pure gas permeation data might be considered as a first estimate of the applicability of membranes to separate a target gas mixture. 

## Figures and Tables

**Figure 1 membranes-11-00833-f001:**
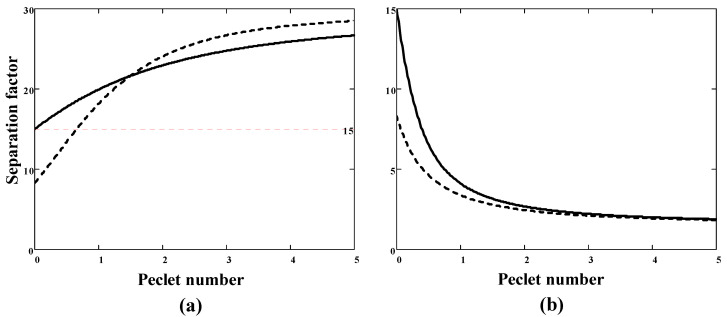
Separation factor for the binary gas mixture (molar fraction $x_{1}=0.1$) with the ideal selectivity $\alpha_{12}=15$ as a function of the membrane Peclét number: (**a**) sorption-selective membrane ($\alpha_{S}=30$, $\alpha_{D}=0.5$); (**b**) diffusion-selective membrane ($\alpha_{S}=1.5$, $\alpha_{D}=10$). Solid lines: the pressure ratio *r* = 0, dotted lines: *r* = 0.1.

**Figure 2 membranes-11-00833-f002:**
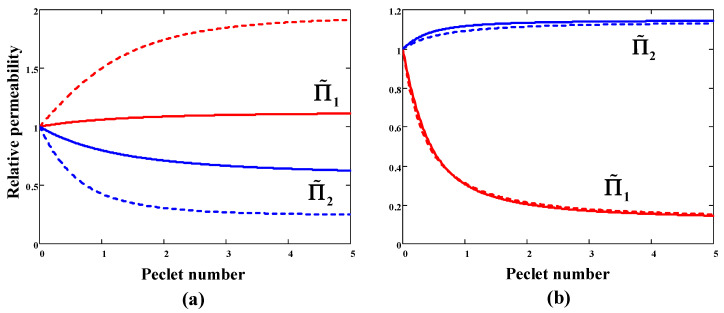
Calculated permeability of component 1 (red lines) and component 2 (blue lines) for the binary gas mixture (molar fraction $x_{1}=0.1$) with the ideal selectivity $\alpha_{12}=15$ as a function of the membrane Peclét number: (**a**) sorption-selective membrane ($\alpha_{S}=30$, $\alpha_{D}=0.5$); (**b**) diffusion-selective membrane ($\alpha_{S}=1.5$, $\alpha_{D}=10$). Solid lines: the pressure ratio *r* = 0, dotted lines: *r* = 0.1.

**Figure 3 membranes-11-00833-f003:**
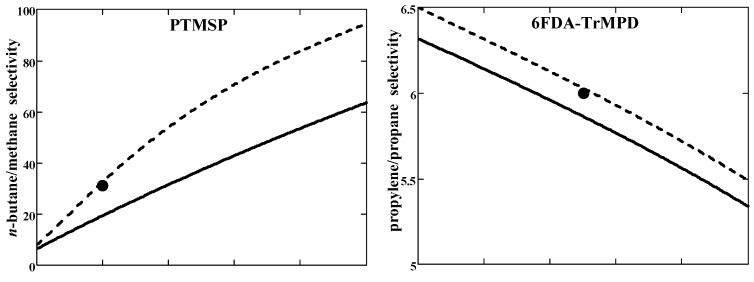
(**a**) Mixed-gas *n*-C_4_H_10_/CH_4_ selectivity and relative permeabilities ${\tilde{\Pi }}_{i}=\Pi_{i}^{mix}/\Pi_{i}^{0}$ of the mixture components through PTMSP membrane versus *n*-C_4_H_10_ feed molar fraction; (**b**) Mixed-gas C_3_H_6_/C_3_H_8_ selectivity and relative permeabilities of the mixture components through polyimide 6FDA-TrMPD versus C_3_H_6_ feed molar fraction. The results of selectivity and permeability calculations are given for pressure ratio *r* = 0 (solid lines) and *r* = 0.05 (dashed lines). Experimental selectivities and permeabilities for 2 mol % *n*-C_4_H_10_/98 mol % CH_4_/PTMSP and 50 mol % C_3_H_6_/50 mol % C_3_H_8_/6FDA-TrMPD systems are shown by the symbols.

**Table 1 membranes-11-00833-t001:** Permeabilities and solubility coefficients (Henry’s constants), and their ratio for *n*-C_4_H_10_/CH_4_ (2 mol %/98 mol %) and C_3_H_6_/C_3_H_8_ (equimolar mixture) in PTMSP and 6FDA-TrMPD membranes, respectively. Diffusivity selectivities are shown in the last column.

Polymer	$\Pi_{1}$ (barrer)	$S_{1}$ (cm^3^(STP) cm^−3^·atm^−1^)	$\Pi_{1}/\Pi_{2}$	$S_{1}/S_{2}$	$D_{1}/D_{2}$
	*n*-C_4_H_10_		*n*-C_4_H_10_/CH_4_		
PTMSP ^1^					
pure	49,000	1156	3.5	286.1	0.012
mixed	68,000	-	31	-	-
	C_3_H_6_		C_3_H_6_/C_3_H_8_		
6FDA-TrMPD ^2^					
pure	30	17.5	11	1.2	8.8
mixed	20	-	6.0	-	-

^1^ Permeabilities from Sultanov et al. [32], solubility coefficients calculated from Raharjo et al. [33] (*T* = 25 °C); ^2^ Data from Tanaka et al. [34], *T* = 50 °C.

## Data Availability

Not applicable.

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
