# Peer review of "Mixed-Gas Selectivity Based on Pure Gas Permeation Measurements: An Approximate Model"

_membranes, 2021, doi:10.3390/membranes11110833_

Round 1
Reviewer 1 Report
The manuscript entitled “Mixed-Gas Selectivity Based on Pure Gas Permeation Measurements: An Approximate Model” the development/modification of an approximate model using Kedem’s friction-coefficient approach for mixed-gas selectivity and permeability of non-porous and microporous polymeric membranes.
Following suggestions need to be incorporated in its revision.
- Grammatical and errors needs to be corrected throughout the manuscript.
- Manuscript needs attentions of native English speaker or help from software for its grammatical and sentence corrections and improvements, which the manuscript lacks significantly.
- Abstracts need to be re-written and grammatical mistakes need to be corrected. More concise abstract can be done.
- Avoid using “We”/’ “Our” and similar words throughout the manuscript. It seems that the authors didn’t consider language throughout the manuscript, which needs special attention. This is turning into major weakness of the manuscript.
- References of the governed equations and related equations are missing throughout the manuscript. It must be provided.
- Under Subheading “ 4.1. Approximations and Flux Equations” , which is assumed to be analogy-based permeability, it must also be mentioned from where it has been used. The basic equation is missing.
- Conclusion must also be rewritten to avoid errors and mistakes.
- Appendix A and B must be provided in supplementary sections rather than in main manuscript.
- Acknowledgement section is missing.
Author Response
Dear Reviewer,
Thank you very much for your comments and suggestions that allowed us, hopefully, to further improve our manuscript. Please find our replay below and we hope that the revised version will address all major concerns and suggestions. The corrections are highlighted in green.
Reviewer #1 comments:
- Grammatical and errors needs to be corrected throughout the manuscript.
- Manuscript needs attentions of native English speaker or help from software for its grammatical and sentence corrections and improvements, which the manuscript lacks significantly.
We did our best to improve the text.
- Abstracts need to be re-written and grammatical mistakes need to be corrected. More concise abstract can be done.
The Abstract has been rewritten according to the aforementioned comments. On the other hand, the missing major results of the work have been added to the Abstract in accordance with comments of the Reviewer #2.
- Avoid using “We”/’ “Our” and similar words throughout the manuscript. It seems that the authors didn’t consider language throughout the manuscript, which needs special attention. This is turning into major weakness of the manuscript.
"We/Our" have been deleted from the text.
- References of the governed equations and related equations are missing throughout the manuscript. It must be provided.
The numbering of formulas and references to equations have been checked and the corresponding corrections have been made.
- Under Subheading “ 4.1. Approximations and Flux Equations” , which is assumed to be analogy-based permeability, it must also be mentioned from where it has been used. The basic equation is missing.
Indeed, the description at the beginning of Sec. 4.1. needs improvement. An additional equation for the permeability coefficients just before Eq. (18) for a more consistent presentation has been inserted.
- Conclusion must also be rewritten to avoid errors and mistakes.
The Conclusion has been rewritten, as well as errors and mistakes have been corrected.
- Appendix A and B must be provided in supplementary sections rather than in main manuscript.
We kindly request the reviewer to agree to leave Appendix A and B in the main body of the manuscript to make the text more reader-friendly.
- Acknowledgement section is missing.
At the end of the text there is a Funding section indicating the appropriate financial support of the study. The "Acknowledgments" is the optional section, which is not required for this article.

Reviewer 2 Report
The manuscript entitled “Mixed-Gas Selectivity Based on Pure Gas Permeation Measurements: An Approximate Model” has been investigated in detail. The topic addressed in the manuscript is potentially interesting and the manuscript contains some practical meanings, however, there are some issues which should be addressed by the authors:
This study may be proposed for publication if it is addressed in the specified problems.
- In the first place, I would encourage the authors to extend the abstract more with the key results. As it is, the abstract is a little thin and does not quite convey the interesting results that follow in the main paper. The "Abstract" section can be made much more impressive by highlighting your contributions. The contribution of the study should be explained simply and clearly.
- The readability and presentation of the study should be further improved. The paper suffers from language problems. The paper should be proofread by a native speaker or a proofreading agent.
- The Introduction section needs a major revision in terms of providing more accurate and informative literature review and the pros and cons of the available approaches and how the proposed method is different comparatively. Also, the motivation and contribution should be stated more clearly.
- The importance of the design carried out in this manuscript can be explained better than other important studies published in this field. I recommend the authors to review other recently developed works.
- The performance of the proposed method should be better analyzed, commented and visualized in the experimental section.
- What makes the proposed method suitable for this unique task? What new development to the proposed method have the authors added (compared to the existing approaches)? These points should be clarified.
- "Discussion" section should be edited in a more highlighting, argumentative way. The author should analysis the reason why the tested results is achieved.
- The authors should clearly emphasize the contribution of the study. Please note that the up-to-date of references will contribute to the up-to-date of your manuscript. The studies named- “Statistical Analysis and Optimal Design of Polymer Inclusion Membrane for Water Treatment by Co(II) Removal”, Desalination and Water Treatment, 182 (2020) 194-207: “Porous Medium Equation in Graphene Oxide Membrane: Nonlinear Dependence of Permeability on Pressure Gradient Explained” Membranes 2021, 11(9), 665. “Polymer Inclusion Membranes with Dinonylnaphthalene Sulfonic Acid as Ion Carrier for Co(II) Transport from Model Solutions”, Desalination and Water Treatment.144 (2019) 185-200- can be used for better explanation of this study.
Author Response
Dear Reviewer,
Thank you very much for your comments and suggestions that allowed us, hopefully, to further improve our manuscript. Please find our replay below and we hope that the revised version will address all major concerns and suggestions. The corrections are highlighted in green.
Reviewer #2 comments:
- In the first place, I would encourage the authors to extend the abstract more with the key results. As it is, the abstract is a little thin and does not quite convey the interesting results that follow in the main paper. The "Abstract" section can be made much more impressive by highlighting your contributions. The contribution of the study should be explained simply and clearly.
The Abstract has been rewritten. The missing major results of the work have been added to the Abstract.
- The readability and presentation of the study should be further improved. The paper suffers from language problems. The paper should be proofread by a native speaker or a proofreading agent.
We did our best to improve the text. We have corrected unclear sentences, mistakes in spelling, grammatical irregularities. The revised text fragments are highlighted in green.
- The Introduction section needs a major revision in terms of providing more accurate and informative literature review and the pros and cons of the available approaches and how the proposed method is different comparatively. Also, the motivation and contribution should be stated more clearly.
The Introduction has been substantially rewritten, new additional recent references were added and discussed:
“Polymeric membranes have been widely used in various gas separation applications mainly due to their high performance regarding permeation and selectivity, as well as the easy processability of polymers. Gas permeation through a dense polymeric membrane is governed by solution-diffusion mechanism where gas dissolution on feed side and diffusion across the membranes determine an overall gas separation process. Although, single gas permeation properties are more often reported, the selectivity of membranes for mixed gas can be different compared to the ideal selectivity based on single gas measurements [1-9]. In some cases ideal selectivities are higher than mixed gas selectivities. For example, CO2/N2 selectivities based on single gas measurement might be higher than mixed-gas selectivities due to the swelling of the polymer in CO2 environment and plasticization effect [10]. The same tendency was observed for hydrocarbons separation through polyalkylmethylsiloxanes composite membranes [9]. The separation of an eight component mixture of saturated and unsaturated hydrocarbons C1-C4 was studied. It was shown that the values of ideal selectivities for C3H8/CH4 and n-C4H10/CH4 gas pairs were higher than mixed-gas selectivities. This effect was explained by significant swelling of the membrane material in the hydrocarbons mixture, first of all, due to the presence of n-butane.
On the other hand, an opposite effect of increase of the separation selectivity for gas mixtures is a character feature of high free volume glassy polymers (so-called polymers of intrinsic microporousity) [1,7]. This unique effect was first reported for poly(1-trimethylsilyl-1-propyne) (PTMSP) by Srinivasan et. al. [11]. It was shown that helium and nitrogen permeation was drastically reduced in the presence of a more strongly sorbing and/or more condensable SF6 in the feed; the phenomena was called as “pore blocking” or “light gas rejection” effect. “
The overwhelming part of publications report the transport properties of polymers based on single gas permeability measurements using “conventional “ barometric or volumetric techniques. The following wording has been added into the text of Introduction: “It should be particularly emphasized that mixed gas measurements need to modify and improve conventional and widely used experimental techniques [18,19].” Besides, the following sentence was added into the Conclusion: “Thus, the model evaluation of mixed-gas selectivity using pure gas permeation data might be considered as a first estimate of the applicability of membranes to separate a target gas mixture.”
The author’s vision of contribution is presented in the last paragraph: “In this work, an attempt was made to develop a simplified analytical model for evaluation and possibly prediction of mixed-gas transport properties of polymer membranes based on available experimental data on single gases. The basic idea was to split the local permeability into two contributions, one of which depends on the friction between the penetrants, and the other contribution can be identified with pure gas permeability.”
- The importance of the design carried out in this manuscript can be explained better than other important studies published in this field. I recommend the authors to review other recently developed works.
The following new publications have been added and discussed in the Introduction: Grushevenko, E.A.; Borisov, I.L.; Volkov, A.V. High-Selectivity Polysiloxane Membranes for Gases and Liquids Separation (A Review). Petrol. Chem. 2021, 61, 959-976: Balҫık, M.; Tantekin-Ersolmaz, S.B.; Pinnau, I.; Ahunbay, M.G. CO2/CH4 mixed-gas separation in PIM-1 at high pressures: Bridging atomistic simulations with process modeling. J. Membr. Sci. 2021, 640, 119838: Yi, S.; Ghanem, B.; Liu, Y.; Pinnau, I.; Koros, W.J. Ultraselective glassy polymer membranes with unprecedented performance for energy-efficient sour gas separation. Sci. Adv. 2019, 5, eaaw5459; Zhang, C.; Fu, L.; Tian, Z.; Cao, B.; Li, P. Post-crosslinking of triptycene-based Tröger's base polymers with enhanced natural gas separation performance. J. Membr. Sci. 2018, 556, 277–284; Sutrisna, P.D.; Hou, J.; Zulkifli, M.Y.; Li, H.; Zhang, Y.; Liang, W.; D'Alessandro, D.M.; Chen, V. Surface functionalized UiO-66/Pebax-based ultrathin composite hollow fiber gas separation membranes. J. Mater. Chem. A 2018, 6, 918–931; Vaughn, J.T.; Harrigan, D.J.; Sundell, B.J.; Lawrence III, J.A.; Yang, J. Reverse selective glassy polymers for C3+ hydrocarbon recovery from natural gas. J. Membr. Sci. 2017, 522, 68–76; Iyer, G.M.; Liu, L.; Zhang, C. Hydrocarbon separations by glassy polymer membranes. J. Polym. Sci. 2020, 58, 2482–2517; Grushevenko, E.A.; Borisov, I.L.; Knyazeva, A.A.; Volkov, V.V.; Volkov, A.V. Polyalkylmethylsiloxanes composite membranes for hydrocarbon/methane separation: Eight component mixed-gas permeation properties. Sep. Purif. Technol. 2020, 241, 116696; Liu, L.; Chakma, A.; Feng, X. CO2/N2 separation by poly(ether block amide) thin film hollow fiber composite membranes. Ind. Eng. Res. 2005, 44, 6874–6882.
- The performance of the proposed method should be better analyzed, commented and visualized in the experimental section.
The work, which is described in the manuscript, is completely theoretical. There are no, i.e. does not contain the Experimental section. The Reviewer is probably referring to Sec. 4.5. (Comparison of the Model with Experimental Data). The purpose of Sec. 4.5 is to illustrate the results of the model for two gas mixture/polymer systems for which the necessary experimental data are available. Undoubtedly, there is always room for better formulation of results. We are quite aware of the imperfection of some of the wording in Sec. 4.5. During the preparation of the revised manuscript the text of Sec. 4.5 was further checked and any grammatical and terminological inaccuracies were corrected.
- What makes the proposed method suitable for this unique task? What new development to the proposed method have the authors added (compared to the existing approaches)? These points should be clarified.
In this work, an attempt has been done to solve a technologically important task, namely, to calculate mixed-gas permeation properties of polymeric membranes based on available experimental data on single gases. The novelty of the developed model compared to the existing approaches is formulated in the corrected/extended Abstract and corrected/extended Conclusions. There are: (1) “The central argument of the model is to decompose the inverse local permeability of penetrants into two terms. The first one is the inverse permeability of pure gas, and the second one is proportional to the friction between penetrants.”; (2) “analytical expressions for permeability and selectivity of polymeric membranes in mixed gas conditions were derived.”; (3) “the developed model as a predictive one if pure gas permeability and solubility coefficients are available as input.”; (4) it is shown that “the Peclét number is less or of the order of unity in hydrocarbon separation by glassy polymers”; (5) “In the limit of large Peclét numbers… the mixed-gas selectivity is equal to the value of the ideal solubility selectivity” (see Conclusions for details). The verification of the model is described in Sec. 4.5 (please, pay attention to the answer to comment #7).
- "Discussion" section should be edited in a more highlighting, argumentative way. The author should analysis the reason why the tested results is achieved.
The verification of the model proposed is presented in Fig. 3 (Sec. 4.5) for two illustrative examples: n-C4H10/CH4 separation through PTMSP membranes and C3H6/C3H8 separation through polyimide membranes. The figure is commented in the text: “Generally, one can state a semi-quantitative agreement with the experiment for both penetrants/polymer systems.” … ”the agreement between the calculated and experimental selectivities improves at non-zero permeate pressure”… “Permeability behavior in mixed gas conditions follows the first (second) inequality (37).” Therefore, the logical consistency of the theory was indicated, however, based on the limited experimental data. In addition: “In terms of the developed model, the blocking of methane transport [in PTMSP] by n-butane is a consequence of frictional interaction between penetrants, in other words, the blocking mechanism is a manifestation of coupling between methane and n-butane fluxes.”
- The authors should clearly emphasize the contribution of the study. Please note that the up-to-date of references will contribute to the up-to-date of your manuscript. The studies named- “Statistical Analysis and Optimal Design of Polymer Inclusion Membrane for Water Treatment by Co(II) Removal”, Desalination and Water Treatment, 182 (2020) 194-207: “Porous Medium Equation in Graphene Oxide Membrane: Nonlinear Dependence of Permeability on Pressure Gradient Explained” Membranes 2021, 11(9), 665. “Polymer Inclusion Membranes with Dinonylnaphthalene Sulfonic Acid as Ion Carrier for Co(II) Transport from Model Solutions”, Desalination and Water Treatment.144 (2019) 185-200- can be used for better explanation of this study.
We are grateful for pointing out interesting research in the field of the membrane separation. Two publications in Desalination and Water Treatment are devoted to the important problem of heavy metals removal from the contaminated water using so-called polymer inclusion membranes. These membranes include the base polymer (cellulose triacetate), a plasticizer and the carrier. Based on the results obtained, an optimal membrane composition providing maximal flux combined with maximal value of Co(II) extraction from aqueous solution was found [Desalination and Water Treatment.144 (2019) 185-200]. In the next publication [Desalination and Water Treatment, 182 (2020) 194-207], the authors use an effective 'central composite design' technique to model and optimize the recovery of Co(II) from the contaminated water. Regretfully, both studies belong to the area related to membrane performance in liquid (aqueous) environment which does not match the topic of manuscript: membrane gas separation.
The publication in Membranes [Membranes 2021, 11(9), 665] is the interesting theoretical study on gas permeability of porous membranes with special focus on the problem of nonlinear dependence of gas permeability on pressure difference. Experimentally, the transport of pure gases through porous graphene oxide (inorganic) membranes was studied. Our manuscript deals with the model development for non-porous polymeric membranes for gas separation. Again, this topic is rather far from the topic of our manuscript. Therefore, these three publications did not consider in the revised manuscript.

Reviewer 3 Report
This manuscript is about a very challenging problem, the prediction of mixed gas selectivities in polymer membranes from permeation data of pure gases. The authors’ approach to consider the coupling effect between interdependent transport of mixture components is based on a modification of a successful formulation in pervaporation, using a friction coefficient formalism. They developed further the existing model with a simple but clever idea, splitting the local permeability into two contributions, one depending on friction between the diffusing components and the second identified with pure gas permeability. They further developed the model in order to enable an analytical solution.
The novelty of the work lies in the development of a simplified analytical model for predicting mixed-gas transport properties of polymer membranes based on experimental data on single gases. The mathematical analysis is rigorous, while the manuscript is well and concisely written and should be published. The validation for both mixtures applied to is based on single experimental points, but this is due to scarcity of experimental data for a challenging measurement problem.
Author Response
Dear Reviewer,
We would like to thank you for considering our manuscript.
